# 3-Chymotrypsin-like Protease (3CLpro) of SARS-CoV-2: Validation as a Molecular Target, Proposal of a Novel Catalytic Mechanism, and Inhibitors in Preclinical and Clinical Trials

**DOI:** 10.3390/v16060844

**Published:** 2024-05-24

**Authors:** Vitor Martins de Freitas Amorim, Eduardo Pereira Soares, Anielle Salviano de Almeida Ferrari, Davi Gabriel Salustiano Merighi, Robson Francisco de Souza, Cristiane Rodrigues Guzzo, Anacleto Silva de Souza

**Affiliations:** Department of Microbiology, Institute of Biomedical Sciences, University of São Paulo, São Paulo 5508-900, Brazil; vitormartins@usp.br (V.M.d.F.A.); eduardo_soares@usp.br (E.P.S.); anielle.ferrari@usp.br (A.S.d.A.F.); davism@usp.br (D.G.S.M.); rfsouza@usp.br (R.F.d.S.)

**Keywords:** SARS-CoV-2, 3CLpro, novel mechanism of catalysis, preclinical and clinical trials, triad

## Abstract

Proteases represent common targets in combating infectious diseases, including COVID-19. The 3-chymotrypsin-like protease (3CLpro) is a validated molecular target for COVID-19, and it is key for developing potent and selective inhibitors for inhibiting viral replication of SARS-CoV-2. In this review, we discuss structural relationships and diverse subsites of 3CLpro, shedding light on the pivotal role of dimerization and active site architecture in substrate recognition and catalysis. Our analysis of bioinformatics and other published studies motivated us to investigate a novel catalytic mechanism for the SARS-CoV-2 polyprotein cleavage by 3CLpro, centering on the triad mechanism involving His41-Cys145-Asp187 and its indispensable role in viral replication. Our hypothesis is that Asp187 may participate in modulating the p*K*_a_ of the His41, in which catalytic histidine may act as an acid and/or a base in the catalytic mechanism. Recognizing Asp187 as a crucial component in the catalytic process underscores its significance as a fundamental pharmacophoric element in drug design. Next, we provide an overview of both covalent and non-covalent inhibitors, elucidating advancements in drug development observed in preclinical and clinical trials. By highlighting various chemical classes and their pharmacokinetic profiles, our review aims to guide future research directions toward the development of highly selective inhibitors, underscore the significance of 3CLpro as a validated therapeutic target, and propel the progression of drug candidates through preclinical and clinical phases.

## 1. Introduction

Coronaviruses (CoVs) are single-stranded positive-sense ribonucleic acid (RNA) viruses, which are divided into four subgroups: Alphacoronavirus, Betacoronavirus, Gammacoronavirus, and Deltacoronavirus [1]. They have a single large genome (27–32 kilobases) directly translated by host cells [2]. The hosts of CoVs are vertebrates that range from human beings to birds, generally causing respiratory and gastrointestinal tract disorders [3]. Seven coronaviruses (CoVs) have emerged to infect humans: HCoV-NL63, HCoV-229E, HCoV-HKU1, HCoV-OC43, Middle East respiratory syndrome coronavirus (MERS-CoV), severe acute respiratory syndrome coronavirus (SARS-CoV), and severe acute respiratory syndrome coronavirus 2 (SARS-CoV-2) [4]. Among these, HCoV-NL63, HCoV-229E, HCoV-HKU1, and HCoV-OC43 typically cause seasonal flu with mild-to-moderate respiratory symptoms. MERS-CoV and SARS caused epidemics, while SARS-CoV-2 caused the 2019 pandemic and it is now considered a seasonal respiratory disease. According to the latest release from the International Committee on Taxonomy of Viruses (https://talk.ictvonline.org/, accessed on 20 May 2024), SARS-CoV-2 and SARS-CoV are classified under the genus of β-CoVs, a subgenus of *Sarbecovirus* and specie of *Betacoronavirus pandemicum*.

In late 2019, a novel severe pneumonia was reported in Wuhan, Hubei Province, China [5]. The World Health Organization (WHO) named this disease “coronavirus disease 2019”, or COVID-19, referring to the first cases of this disease reported in 2019 [6]. Until March 2024, SARS-CoV-2 caused a total of 7,033,430 deaths and infected more than 774 million individuals. Since the virus’s emergence, until March 2024, more than 3.9 thousand different genomes have been recorded [7]. The virus accesses the host cell through the Spike protein, which binds to the human ACE2 receptor [6,8,9,10]. The virus may infect human cells by two primary entry pathways: (1) fusion of the virus membrane with the cell membrane, facilitating the release of its genetic material into the cytoplasm of the host cell or/and (2) internalization of the virus via endocytosis. Following these entry mechanisms, the virus proceeds to initiate its replicative process, whereby its genome is released into the cytoplasm of the host cell. Upon entry into the host cell, the viral genome is released and acts as a template for the synthesis of viral proteins and genome replication. The ORFa and ORF1ab genes are then translated into two important polyproteins, respectively, pp1a and pp1ab [11,12,13]. The production of these polyproteins involves overlapping regions that are initiated by a ribosomal frameshift within this region. Collectively, the genome of SARS-CoV-2 encodes four structural proteins, sixteen non-structural proteins (nsps), and nine accessory proteins [14]. Two proteases—papain-like and 3-chymotrypsin-like proteases (PL and 3CLpro, respectively)—cleave the polyprotein [14,15]. The polyproteins are cleaved at least in eleven sites by the SARS-CoV-2 3CLpro [16]. Initially, the 3CLpro releases itself from the polyproteins through autocleavage and, subsequently, it forms a functional homodimer, enabling it to cleave pp1a and pp1ab transversely. However, the exact mechanism by which the 3CLpro carries out the cleavage of pp1a and pp1ab remains unknown.

Given the vital role of these proteases in viral replication, especially during the pandemic caused by SARS-CoV-2, they have emerged as promising targets for the development of broad-spectrum anti-SARS-CoV-2 agents [17]. The absence of human homologs of 3CLpro enables the development of specific inhibitors with minimal side effects on human proteases, potentially reducing the adverse effects associated with these inhibitors [18]. The 3CLpro has sparked significant interest in both the academic field and the pharmaceutical industry as a molecular target to inhibit viral replication and pathogenesis in various coronaviruses [19]. Several drug discovery strategies have been employed to find 3CLpro inhibitors, including drug repurposing, high-throughput virtual screening (HTVS), high-throughput screening (HTS), and structure- and ligand-based drug designs (SBDD and LBBD, respectively) [20,21,22,23,24,25,26]. Additionally, the strategy of exploring natural products as 3CLpro inhibitors has been one of the most important approaches for developing novel anti-SARS-CoV-2 agents [27,28]. These efforts have led to chemically diverse synthetic compounds and natural products emerging as effective inhibitors of SARS-CoV-2 3CLpro [29,30].

Herein, we reviewed the main protease of SARS-CoV-2 as a pivotal and promising therapeutic target for the treatment of COVID-19. We propose that the cleavage of polyproteins pp1a and pp1ab is facilitated by the triad H41-Cys145-Asp187, based on our bioinformatics analyses coupled with computational and experimental data from the literature. This hypothesis assumes critical significance, as Asp187 emerges as a pharmacophoric determinant justifying consideration in the optimization of bioactive compounds targeting the virus’s principal enzyme. Next, we discuss the compounds evaluated both in vitro and in vivo, alongside those advancing through clinical drug development phases. From a pharmacokinetic standpoint, our effort is to correlate the biological attributes with the physicochemical features of the drug-like compound, with the goal of discerning requisite patterns favoring the effective attenuation of viral pathogenesis. Taken together, our study brings important insights into optimizing lead compounds in the preclinical stage of development.

### 1.1. 3-Chymotrypsin-Like Protease: A Validated Molecular Target

A crucial strategy in rational drug design is to target molecular entities that lack homologs in humans. The absence of homolog proteins of 3CLpro in humans is critical because inhibitors must be designed to specifically interact with the viral protease, thereby reducing the probability of affecting human proteases. Homologs are similar proteins found in different species, and the presence of human homologs of 3CLpro could increase the risk of adverse effects during administration in humans. By employing the Basic Local Alignment Search Tool (BLASTP) and Position-Specific Iterated BLAST (PSIBLAST) tools on the National Center for Biotechnology Information (NCBI, https://www.ncbi.nlm.nih.gov/, accessed on 20 May 2024), considering the sequence of 3CLpro from SARS-CoV-2 (PDB ID 7WOF) [31], we examined the sequence of 3CLpro from SARS-CoV-2 and restricted the search only to *Homo sapiens* (taxid:9606). Although we did not find human proteins with sequence similarity to 3CLpro of SARS-CoV-2, it is possible that some human proteins share structural similarities. In this regard, we also searched the Dali server [32,33] using the structure of the 3CLpro A chain (PDB ID 7EN8) [34] as a query to search for distant homologs in humans. The results of the search for structural homologs of 3CLpro in human proteins, obtained from the Dali server, are available in Appendix A. We found serine protease Fam111a, prothrombin, serine protease hepsin, mannan-binding lectin serine protease, complement C3b beta chain, enteropeptidase, protein-Z-dependent protease inhibitor, human complement factor I, coagulation factor XI, and plasminogen, in which root-mean-square deviation (RMSD) ranged from 2.7 to 3.5 Å, with a similarity of amino acid sequences between 8 and 16% and a Z-score between 9.1 and 11.5 (Appendix A). However, the significant discrepancy in sequence homology between 3CLpro and identified human proteases suggests that 3CLpro inhibitors may not be effective against human proteases.

In addition to identifying possible homologs of 3CLpro in humans, it is crucial to verify the specificity of this molecular target, ensuring its essentiality in viral replication. One approach to demonstrate this is the inhibition of 3CLpro in several strains of coronavirus. By validating this essentiality, it is possible to extrapolate its importance to all viral lineages, consolidating it as a fundamental molecular target for drug development. Yang and co-authors were pioneers in identifying molecular targets for the development of broad-spectrum inhibitors against coronavirus diseases [35]. They carried out an initial screening of targets, excluding structural proteins due to notable variations between different strains of coronavirus [35]. After the selection, they focused on RNA-dependent RNA polymerase (RdRp), RNA helicase, and Mpro as possible targets among the non-structural proteins [35]. Despite the differences in the primary sequence of Mpro between coronaviruses, a three-dimensional analysis revealed a high similarity, especially in the active site [35]. Based on that, Yang and co-authors developed irreversible inhibitors, demonstrating efficacy against several Mpros of different coronaviruses, including SARS-CoV, MERS-CoV, and others [35], showing that 3CLpro is a key protease for replication of the coronavirus.

### 1.2. Structure and Function

Mpro^SARS-CoV−2^ is a cysteine protease [36] sharing 97% sequence similarity with SARS-CoV [37]. It demonstrates comparable catalytic behavior [19]. Structurally, SARS-CoV-2 Mpro exists as a dimer, with a dissociation constant (*K*_D_) in the micromolar range [36]. Its three-dimensional structure includes domains I and II (residues 10–99 and 100–182, respectively), featuring six barrel structures arranged into antiparallel β-strands (Figure 1a,b) [38]. Conversely, domain III (residues 198–303) comprises five α-helices governing 3CLpro dimerization [39]. Mutation within domain III leads to a monomeric Mpro and therefore results in catalytic activity loss [40,41,42]. The enzyme’s active site is localized between domains I and II, in which it has specific subsites (S1’, S1, S2, S3/S4’) (Figure 1c,d) [43,44,45]. Notably, catalytic histidine, H41 residue, occupies the catalytic S1’ subsite, while residues Cys145, His172, Glu166, His163, His164, and Phe140 occupy the S1 subsite. Hydrophobic residues such as Met49, Tyr54, Met165, Pro168, and Val186 belong to the S2 subsite, engaging with the substrate’s hydrophobic segment at the P2 position [38]. Subsites S3 and S4, consisting of residues Gln189, Gly251, Gln192, and Ala191, remain more solvent-exposed. The limited catalytic activity of monomeric Mpro underscores the importance of dimerization in substrate recognition, especially at the S1 subsite. Consequently, several inhibitors of 3CLpro have been developed to target protein dimerization.

The structure and function of 3CLpro are crucial for understanding the processing of viral polyproteins [48]. Since the active site of SARS-CoV-2 3CLpro is broad and contains subsites with different physicochemical features, the substrate binds in a manner that occupies the entire active site [49]. Positions P1, P2, and P1ʹ on the substrate play a crucial role in enzymatic specificity, while positions P3 and P4 expand the interaction area and stabilize the substrate [50]. Zhao and colleagues demonstrated the sequence alignment of residues surrounding eleven cleavage sites, revealing a specific pattern within the polyprotein for substrate recognition by the enzyme. This pattern includes P4 (A/V/P/T), P3 (T/K/R/V/M), P2 (L, highly conserved/F/V), P1 (conserved for Q), and P1’ (S, reasonably frequent/A/N) [16]. Therefore, these data demonstrate that, for certain positions, 3CLpro keeps specificity for this pattern.

### 1.3. A Novel Proposal for the Mechanism of Polyprotein Processing

In this section, we present a novel proposal for the cleavage mechanism of the SARS-CoV-2 polyprotein. Our analysis is carried out by integrating our bioinformatics data with experimental and computational data reported in the scientific literature. Initially, we begin the discussion with an analysis of the catalysis mechanism involving the His41/Cys145 dyad, elucidating how these catalytic residues participate in the cleavage process and which is currently accepted as a catalysis mechanism. Subsequently, we propose a hypothesis that the cleavage of the SARS-CoV-2 polyprotein may be intrinsically related to the coordinated action of the His41/Cys145/Asp187 catalytic triad.

The currently accepted catalytic mechanism involves the His41/Cys145 dyad in polyprotein cleavage [51]. Typically, position P1 is occupied by glutamine [16,50], which is an almost universal requirement [16]. The catalytic dyad is activated by water and maintained by residue Asp187 [52]. Once activated, the catalytic cysteine acts as a nucleophile in the proteolytic process, with its activation occurring through deprotonation of the thiol group by His41 [53]. The resulting thiolate then undergoes nucleophilic attack on the substrate’s glutamine, resulting in amide bond cleavage. Subsequently, the catalytic histidine restores its deprotonated form, allowing the thioester to be attacked, with water acting as a nucleophile and releasing the hydrolyzed C-terminal, restoring the catalytic dyad [52].

It is known that chymotrypsin, a homolog of 3CLpro, has a catalytic mechanism containing a catalytic triad Asp102-His57-Ser195 [54]. As described earlier, SARS-CoV-2 3CLpro has a mechanism containing a dyad. We searched published articles, site-directed mutagenesis studies, and computational and biochemical studies that demonstrate the possibility of a catalytic triad mechanism participating in the catalysis process.

In order to search for the amino acid residues of 3CLpro from different lineages of each coronavirus genus, we initially used BLASTP (Basic Local Alignment Search Tool) to locate representatives of each genus, while excluding SARS-CoV-2 3CLpro sequences at all stages (as shown in Appendix A). This resulted in fifteen sequences from different coronavirus lineages, with an additional 3CLpro sequence from SARS-CoV-2 included, totaling sixteen sequences (as shown in Appendix A). Subsequently, we conducted a multiple sequence alignment of these amino acid sequences using Clustal Omega (https://www.ebi.ac.uk/jdispatcher/msa/clustalo, accessed on 20 May 2024) with default parameters (Appendix A). Finally, the alignment was processed through the WebLogo site (https://weblogo.berkeley.edu/logo.cgi, accessed on 20 May 2024) to generate the sequence logo (as illustrated in Appendix A). Appendix A shows the absolute conservation of His45, Cys145, and Asp187 among 3CLpro sequences from diverse coronavirus lineages, consistent with previous studies [39,55,56,57]. Our bioinformatics investigations have highlighted a range of conserved amino acid residues in 3CLpro, some associated with dimerization and others with the catalytic process of polyprotein cleavage. These findings are in agreement with published studies [55,56,57,58]. However, our focus was on the aspartate residue within the His41, Cys145, and Asp187 catalytic triad.

Our previous molecular dynamics study showed that Asp187 played an important role in interacting with the catalytic histidine in different 3CLpro/substrate complexes [39], presenting extremely significant hydrogen bonds (hydrogen bonding occupancy above 50%) [39]. In our previous study on the identification of novel 3CLpro inhibitors, we observed that the contribution of the binding affinity of the Asp187 was comparable to Glu166, suggesting that both are critical for the investigation of novel inhibitors [23]. Other computational simulation works have corroborated the hypothesis that Asp187 is crucial in interacting with the catalytic histidine [59]. A quantum mechanics–molecular mechanics (QM-MM) study seeking to understand the mechanism of action of the PF-0721332 inhibitor on SARS-CoV-2 3CLpro described that Asp187 was close to the His41 residue and could form a catalytic triad, Cys145–His41–Asp187, which would facilitate the deprotonation of Cys145 [59]. In this regard, Asp187 was included in the QM region, and it was verified that in this covalent inhibition mechanism between Mpro and PF-0721332, there was a strong hydrogen bond (distance of 1.64 Å between His41 and Asp187) [59]. These results show that the catalytic triad Cys145–His41–Asp187 plays an important role in the covalent inhibition of SARS-CoV-2 Mpro, allowing the deprotonation of Cys145 and thus facilitating subsequent reaction (Figure 2).

Additionally, Adem and co-authors performed a series of experiments using site-directed mutations in the main residues of the different subsites of 3CLpro, comparing the proteolytic activity of the mutants relative to 3CLpro^WT^. The alanine substitution in D187 (3CLpro^D187A)^ abolished 3CLpro activity [60]. Furthermore, the D187E and D187N substitutions resulted in 3CLpro inactivation [60]. Taken together, all results combinated suggest that the catalytic mechanism of 3CLpro is formed by a His41-Cys145-Asp187 triad. Indeed, the proposal of the catalytic triad participating in reactions is classic in several catalytic reactions of various enzyme classes [61,62,63,64,65]. For example, QM-MM studies conducted by Arafet and co-authors showed that the residues Gln19, Asn175, and Trp181, located near His159 of the cruzain (the major protease of *Trypanosoma cruzi*), are essential for modulating the p*K*_a_ of the catalytic histidine, allowing it to act as acid and base in the catalysis [66]. This motivated us to calculate the p*K*_a_ of Asp187 using the structure of 3CLpro (PDB 7WOF) and submit it to the ProPka (https://www.ddl.unimi.it/vegaol/propka_run.php, accessed on 20 may 2024) [67,68]. Interestingly, we found that Asp187 has an extremely low p*K*_a_ (p*K*_a_~−0.7), making it an extremely strong acidic. In this regard, our hypothesis for 3CLpro is that Asp187 may participate in modulating the p*K*_a_ of His41, allowing His41 to act as a base in the acylation and deacylation phases. Therefore, the proposed mechanism in this review follows two phases, acylation followed by deacylation, where the roles of these residues would be analogous to the catalytic triad of 3CLpro (Figure 2c).

*I. Acylation step.* Initially, His41 acts as a base, where Asp187 may participate in modulating the p*K*_a_ of His41, taking a proton from Cys145 and activating it as a nucleophile. Then, the nucleophilic Cys145 attacks the carbon of the carbonyl group of the main chain of the glutamine of the substrate, forming a tetrahedral intermediate. This leads to cleavage of the peptide bond, releasing a substrate fragment with an alcohol group and forming an acyl bond between Cys145 and the other fragment of the substrate (Figure 2c).

*II. Deacylation step.* In this step, a water molecule acts as a nucleophile by a hydrolysis reaction. His41 now acts as a base, where Asp187 may participate in modulating the p*K*_a_ of His41, taking a proton from the water molecule and activating the water to perform the nucleophilic attack on the acyl group in Cys145, forming a tetrahedral intermediate. This results in the cleavage of the acyl bond to form a carboxylic acid and the regeneration of the active form of the main protease. The second fragment of the substrate is released, completing the reaction (Figure 2c).

This catalytic mechanism, adapted for the main protease of SARS-CoV-2 and other coronavirus lineages, is crucial for the processing of viral polyproteins. As a result, we suggested Asp187 as a pharmacophore in the design of 3CLpro inhibitors, where inhibitors that interact with this residue destabilize the hydrogen bond interaction between the residue pair His41/Asp187, affecting the modulation of the p*K*_a_ of His41 in the catalysis mechanism.

### 1.4. Preclinical and Clinical Trials of Reversible and Irreversible 3CLpro Inhibitors against SARS-CoV-2

The compounds tested on the Mpro originated from various sources such as peptides, non-peptides, chemicals, and flavonoid derivatives, among others [50,69,70,71]. Most are competitive inhibitors focused on allosteric inhibition [72]. The global emergence of COVID-19 led to the fast repurposing of protease inhibitors previously developed for other viral indications [73]. Although HIV protease inhibitors did not provide any benefit for COVID-19 treatment [74], it was the first-generation peptidomimetic protease inhibitors that were initially tested against the main enzyme of SARS-CoV-2. These molecules have a Gln substitution at the P1 position but do not exhibit significant inhibitory activity against SARS-CoV-2’s 3CLpro (for example, see reference [75]). Second-generation Mpro inhibitors showed good in vitro and in vivo activity in animal models against MERS and in vitro activity against SARS-CoV-2 [76]. Although the SARS-CoV-2 primarily infects the respiratory tract (pharynx, trachea, lungs) in the early stages of infection, a broader organotropism has been reported in more severe and advanced cases of the disease [77]. Herein, we compared the pharmacokinetic performance of the 3CLpro inhibitors with Lipinski’s rules and topological polar surface area (TPSA). Lipinski’s rule, also known as the “Rule of Five”, assesses the viability of a substance for oral administration [78,79]. The rules are based on physicochemical properties that favorably influence the absorption and permeability of a molecule in the human body, and they are as follows: (1) the molecule should have a molecular weight of 500 Da or less; (2) Log *P* (octanol/water partition coefficient) should be 5 or less, indicating balanced solubility in water and lipids; (3) the number of hydrogen bond donors in the molecule should be no more than 5 (usually NH or OH groups); and (4) the number of hydrogen bond acceptors in the molecule should be no more than 10 (usually nitrogen or oxygen atoms) [78,79]. We also used TPSA as a parameter since it measures the ability of a molecule to permeate cells. In general, molecules with a TPSA greater than 140 Å^2^ tend to have poor permeability across cell membranes [80]. The combination of all these parameters may help understand why some drugs are successful while others are not in the preclinical and clinical trials.

CDI-45205 (structure unavailable) is administered by intravenous or inhalation [81,82]. Other 3CLpro inhibitors have also been administered intravenously or intraperitoneally to increase oral bioavailability. PF-00835231 and GC376 inhibitors presented IC_50_ values of 0.004 and 0.2 μM and EC_50_ values of 0.2 and 2.2 μM, respectively (Table 1). Both presented low oral bioavailability in rats (1.4% and 3%, respectively) [83]. PF-00835231 and GC376 are examples that have TPSA values of 149.6 and 182.3 Å^2^, both greater than 140 Å^2^. In general, molecules with values above 140 Å^2^ are considered difficult to permeate through membranes [80], which may result in low bioavailability in the circulatory system. Conversely, pharmacodynamics optimizations of PF-00835231 led to the PF-07304814 [84] (Figure 3 and Table 1).

In general, second-generation inhibitors aim to enhance pharmacokinetic properties, particularly by increasing oral bioavailability [82]. For instance, peptidomimetic inhibitors targeting the human rhinovirus 3-chymotrypsin protease have demonstrated oral bioavailability exceeding 20% in rodent and other nonclinical models, suggesting their potential suitability for oral administration [82]. Pfizer has initiated a Phase II clinical trial with the second-generation oral 3CLpro inhibitor PF-07321332 (Nirmatrelvir) (Table 1 and Table 2), which has been specifically optimized for oral administration [91]. This inhibitor features a nitrile head and has undergone optimization due to a reduction in hydrogen bond donors and the incorporation of a trifluoroacetyl group, which presented a *K*_i_ value of 3 nM for Mpro and presented a high antiviral activity in Vero E6 cells infected with SARS-CoV-2, with EC_50_ value of 0.08 μM with good selectivity and safety in vivo [87,88] (Table 1). The National Medical Products Administration (NMPA) approved the SIM0417 SARS-CoV-2 Mpro inhibitor for the treatment of adults with mild to moderate COVID-19 symptoms on 28 January 2023. However, the structure of SIM0417 has not been disclosed. Similar to Nirmatrelvir, SIM0417 requires low-dose ritonavir administration, which slows down first-pass metabolism in vivo and enhances efficacy in inhibiting the virus life cycle [95] (Table 2).

Nirmatrelvir was developed by Pfizer in 2002 for the treatment of SARS-CoV; this pharmaceutical company backed studies in 2021 to evaluate its efficacy against SARS-CoV-2 [96,97]. The drug exhibits in vitro and in vivo antiviral activity, and currently, the study is in Phase II clinical trials to assess its safety, tolerability, and pharmacokinetics (Table 1). However, Mpro SARS-CoV-2 variants containing mutations in the E166N/V, M165T, G143S, Q189E, A173V, H172F/Q/Y, or Q192S/T/V residues render viruses resistant to Nirmatrelvir treatment [98]. Recently approved by the FDA, on 22 December 2021, Paxlovid is a novel oral medication that combines a 3CLpro inhibitor (Nirmatrelvir) with Ritonavir (Pfizer PF-07321332/ritonavir) to treat patients with moderate or severe COVID-19. Nirmatrelvir inhibited 3CLpro from several types of human coronaviruses, besides possessing antiviral activity in Vero E6 cells with an EC_50_ value of 74.5 nM. The clinical trial of phase II/III with Nirmatrelvir and ritonavir in non-hospitalized COVID-19 adults reduced the risk of hospitalization or death by 89% [99]. In the Omicron BA.2 subvariant, Paxlovid treatment decreased COVID-19 progression and mortality due to viral load reduction [100].

Boceprevir is used as a protease inhibitor for the hepatitis C virus (HCV) [101] by containing an α-ketoamide that forms a covalent bond with the serine active site in the 3CLpro protein. It proved to be an effective drug in virtual screening and in vitro assays. Boceprevir, in vitro, showed effective inhibition, with an IC_50_ of 4.1 μM, and the EC_50_ value was 1.3 μM. However, more in vitro and in vivo studies are necessary in the preclinical studies. Boceprevir violates Lipinski’s rules due to its molecular weight (519.7 Da). Furthermore, we observed that Boceprevir has a TPSA value of 150.7 Å^2^ (more than 140 Å^2^, which makes it difficult to cross through cell membranes). Comparing Nirmatrelvir with Boceprevir, both present the same chemical scaffold (Table 1). However, Boceprevir has the chemical groups cyclobutane, isopropyl urea, and oxopropyl amine group in the same position as the pyrrolidinone, trifluoromethyl amide, and nitrile groups in Nirmatrelvir, respectively. Analyzing Table 1, we observed that Nirmatrelvir has more favorable pharmacokinetic parameters (MW = 499.5 Da; TPSA = 131.4 Å^2^; HBA = 8; HBD = 3; Log *P* = 1.9) for crossing biological membranes than Boceprevir (MW = 519.7 Da; TPSA = 150.7 Å^2^; HBA = 5; HBD = 4; Log *P* = 2.1), suggesting therefore that these physicochemical properties may be affecting the pharmacokinetic properties of Boceprevir.

Lopinavir and Ritonavir are HIV protease inhibitors used in AIDS treatment (Table 2) [102,103]. Both were employed in treating SARS-CoV-2 infections due to their ability to reduce SARS-CoV 3CLpro activity in vitro. Despite in vitro studies showing a decrease in viral replication, in vivo studies in animal models demonstrated that the drug combination was ineffective in the SARS-CoV-2 treatment [104]. In 2021, the World Health Organization (WHO) did not recommend the use of these medications for COVID-19 treatment. Despite their withdrawal from clinical trials, Ritonavir has found new utility when combined with Nirmatrelvir, currently undergoing phase II/III clinical trials (Table 2). Ritonavir is used as a combined strategy with Nirmatrelvir because it improves pharmacokinetic properties by inhibiting CYP3A4 [105]. Nirmatrelvir’s rapid metabolism may interfere with the treatment of COVID-19. Ritonavir acts by decreasing the metabolism of Nirmatrelvir by inhibiting CYP3A4 and consequently improving the efficacy of Nirmatrelvir [106]. Furthermore, Ritonavir has a high binding affinity to plasma proteins, suggesting that the increased action of Paxlovid may also be related to the efficacy of this drug [107]. When combined, Ritonavir may occupy the binding sites of these proteins in blood plasma, increasing the free concentration of Nirmatrelvir to have its antiviral action [107].

The α-ketoamides were envisioned as broad-spectrum inhibitors of the main proteases of beta coronaviruses and alpha coronaviruses, as well as enterovirus 3C proteases [91,108]. Compound **11r** showed promise, with an EC_50_ of 400 pM against MERS-CoV in Huh7 cells and EC_50_ against SARS-CoV and a variety of enteroviruses in various cell lines [46]. Building on these results, Zhang and co-authors modified compound **11r** by replacing the amide at the P2-P3 position with a pyridone ring, aiming to enhance selectivity for the SARS-CoV-2 protease [46] (Figure 4). Another modification was made to improve one of the pharmacokinetic properties, plasma solubility, and reduce plasma protein binding [46]. The authors replaced the *tert*-butyloxycarbonyl protecting group or *tert*-butoxycarbonyl protecting group (Boc) with a slightly less hydrophobic group, resulting in compound **13a** (Figure 4) [46]. The α-ketoamides are a class of reversible inhibitors [109]. 3Clpro Cys145 undergoes a nucleophilic attack on the α-keto group of the α-ketoamide, forming a thiohemiacetal in a reversible reaction [109]. Compound **13b** was able to inhibit the recombinant purified SARS-CoV-2 Mpro with IC_50_ = 0.7 μM. In vitro assays showed that Compound **13b** was effective against SARS-CoV and MERS-CoV, as well as inhibiting SARS-CoV-2 RNA replication in Calu-3 cells (EC_50_~5 μM) [46]. To enhance pharmacodynamic properties, the authors substituted the cyclohexyl group at the P2 position with a cyclopropyl in Compound **13b**. The absence of the Boc group in this class of inhibitors is necessary for crossing the cell membrane but leads to higher plasma protein binding, a disadvantage in pharmacokinetic properties [46]. Compound **11r** effectively inhibited MERS-CoV and SARS-CoV, leading to modifications, as present in compound **13a**, to enhance selectivity for SARS-CoV-2 protease [46]. Pharmacokinetic studies revealed adequate metabolic stability for compound **13a**, with primarily tissue distribution and excretion via urine [46]. Compound **13b** exhibited similar properties and pulmonary tropism, suggesting potential efficacy in the treatment of COVID-19 [46]. Inhalation administration was well tolerated, indicating the possibility of direct lung delivery [46]. Therefore, α-ketoamides represent a promising class of inhibitors for viral diseases, with compounds **13a** and **13b** showing potential in both in vitro studies and animal models.

Carbonic groups such as aldehydes and ketones stand out as promising targets for the development of novel covalent inhibitors of SARS-CoV-2 Mpro [110,111,112]. Generally, the formation of the covalent bond depends on the electrophilic nature of the carbonyl carbon, which is susceptible to nucleophilic attack by the catalytic cysteine, resulting in the formation of a reversible thiohemiketal adduct [113]. The structural similarity between this adduct and the intermediate formed by the natural substrate during the enzymatic catalytic cycle ensures the high stability of the inhibitor–protein complex and prolonged residence time [113]. Examples of these covalent inhibitors, such as compounds **1** and **2** (Table 1), have been derived as peptidomimetics and have shown excellent inhibitory activity against SARS-CoV-2 Mpro (IC_50_ = 0.05 μM and 0.04 μM, respectively) and antiviral activity against SARS-CoV-2 (EC_50_ values of 0.5 and 0.7 μM, respectively) [30]. Moreover, these compounds have demonstrated low cytotoxicity and favorable pharmacokinetic and toxicological properties in vivo. These data suggest that compounds **1** and **2** are promising candidates to advance to clinical trial phases in the treatment of COVID-19.

Ensitrelvir (S-217622), a non-covalent inhibitor of the 3CLpro, was evaluated using Vero E6T cells against eleven SARS-CoV-2 variants, including eight Omicron variants [114]. It exhibited consistent antiviral activity across all tested SARS-CoV-2 variants, with EC_50_ values ranging from 0.2 to 0.5 µM [114] (Table 1 and Table 2). The IC_50_ values of Ensitrelvir against 3CLpro of these variants ranged from 8.0 to 14.4 nM [114], comparable to 3CLpro^WT^ (IC_50_^WT^ = 13 nM [92]). Treatment with Ensitrelvir significantly reduced the SARS-CoV-2 viral titer compared with placebo [115]. Notably, COVID-19 treatment with Ensitrelvir improved respiratory symptoms and fever, which are among the most common symptoms associated with the Omicron variant [115]. However, transient changes in HDL cholesterol, triglycerides, total bilirubin, and iron levels were observed post-treatment, indicating the need for further studies on these effects during Ensitrelvir treatment [115]. The safety and efficacy of Ensitrelvir in a larger population will be evaluated in phase III trials [115]. Overall, once-daily oral administration of Ensitrelvir for 5 days demonstrated antiviral efficacy and had an acceptable safety profile in patients with mild-to-moderate COVID-19, most of whom were vaccinated [115].

**Table 2 viruses-16-00844-t002:** **3CLpro inhibitors in preclinical and clinical trials.** The table provides an overview of 3CLpro inhibitors in preclinical and clinical trials. Each row represents a specific molecule under investigation, with information about the 3CLpro inhibitors’ name, the pharmaceutical company responsible for their development, the route of administration, the current state of development, the inhibitory concentration (IC_50_) in nanomolar for 3CLpro inhibitors in preclinical trials, the effective concentration (EC_50_) in micromolar for 3CLpro inhibitors in preclinical or clinical trials, and the unique identifier (ID) of the registered clinical trial. These data offer a comprehensive perspective on 3CLpro inhibitors in the research and development phase for potential use in treating COVID-19.

Molecule	Company	Delivery	State	IC_50_ (nM)	EC_50_ (μM)	ID (Clinical Trials)
Nirmatrelvir (PF-07321332)	Pfizer	Oral	Phase II	10–100	0.07–0.3	NCT05011513
Lopinavir and Ritonavir	Abbvie	Oral	Discontinued	13,700.0 ^#^	26.6 *	NCT04381936
Paxlovid™ (Nirmatrelvir and Ritonavir)	Pfizer	Oral	Phase II/III	3.1	0.075	NCT04960202
Ensitrelvir (S-217622)	Shionogi	Oral	Phase II/III	13	0.2–0.5	NCT05305547
Boceprevir	Merck	Oral	Preclinical	1.6	2–10 ^&^	ND
GC376	Kansas State University	Oral	Preclinical	26.4	2.6 ± 0.2 **/1.1 ± 0.2 ***	ND
Lufotrelvir (PF-07304814)	Pfizer	IV	Phase I	0.27	760	NCT05050682
Xiannuoxin™: Simnotrelvir (SIM0417) and Ritonavir (SSD8432)	Simcere	Oral	Phase II/III	9	34	NCT05373433 [116]

^#^ IC_50_ determined for Ritonavir; * Vero E6 and A549, overexpressing ACE2/TMPRSS2 cells; ^&^ assays performed in HEK293T/17 cells; ** assays performed in Vero cells; *** assays performed in Calo-2 cells; IV = intravenous administration.

## 2. Conclusions

The 3CLpro protease plays a crucial role in viral replication, characterized by its high degree of protein conservation among members of the *Coronaviridae* family. Its specificity for viral proteins renders it a promising target for drug development aimed at inhibiting viral replication. However, the urgent need for anti-SARS-CoV-2 drugs during the pandemic prompted many countries to approve various 3CLpro inhibitors, some of which were originally developed as antivirals for other diseases [117]. Unfortunately, since 2020, many of these approved drugs have been discontinued due to their ineffectiveness in combating the virus in vivo, as well as their inability to reduce hospitalizations and due to their high toxicity [118]. Additionally, the emergence of SARS-CoV-2 variants has rendered some of these drugs ineffective against the new strains [119]. Hence, the search for novel molecules capable of inhibiting virus replication is paramount, serving as a preventive measure against future coronavirus-induced pandemics. Another significant contribution in this review is the hypothesis regarding Asp187 as a potential pharmacophoric component in the design of inhibitors. We have evidence to suggest that it is part of a catalytic triad involved in processing the virus polyprotein. Including Asp187 as a pharmacophoric element could be crucial in enhancing the activity of bioactive compounds undergoing molecular optimization. We believe that if the catalytic aspartate interacts with a specific inhibitor, it will destabilize the triad, thus impeding proton transfer in the acylation and deacylation stages, which are fundamental for catalysis.

## Figures and Tables

**Figure 1 viruses-16-00844-f001:**
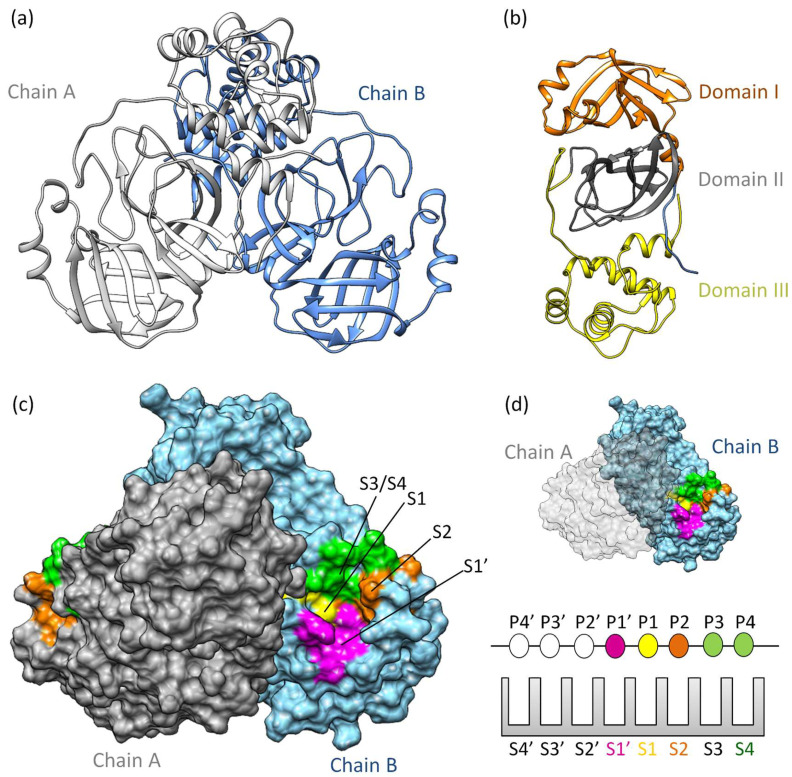
**3CLpro structure and function:** (**a**) The three-dimensional structure of 3CLpro homodimer colored based on chain (PDB ID 7WOF) [31]; (**b**) 3Clpro monomer has three distinct domains: I, II, and III. Domains I (residues 10–99) and II (residues 100–182) have six antiparallel β-strands, forming a stage for the catalytic site [46,47]. Meanwhile, Domain III (residues 198–303) unveils a captivating structure featuring a globular cluster of five α-helices, steering 3CLpro dimerization regulation through salt bridge interactions. (**c**) The substrate binding site of 3CLpro is a composition of the subsites S1, S1’, S2, and S3/S4, (**d**) following the Schecter–Berger nomenclature for proteases. 3CLpro’s active site features S1’, S1, S2, and S3/S4 subsites, with Pis and Pi’s denoting substrate positions. The S1–S1’ interface marks the cleavage site, initiating numbering. X-ray crystal structure studies confirm the subsites S1’, S1, S2, and S3/S4 [44,45]. The S1 subsite (F140, C145, H163, E166) ensures structural stability and substrate recognition, featuring key π–π interactions. S1’ (T25, T26, H41, L27, N142, G143) is exposed to the aqueous environment, accommodating smaller side chains of the substrate. S2 (M49, Y54, P52, H164, M165, D187, R188, Q189) is highly hydrophobic, and S3/S4 (M165, E166, L167, P168, F185, T190, A191) completes the hydrophobic subsite composition [44,45].

**Figure 2 viruses-16-00844-f002:**
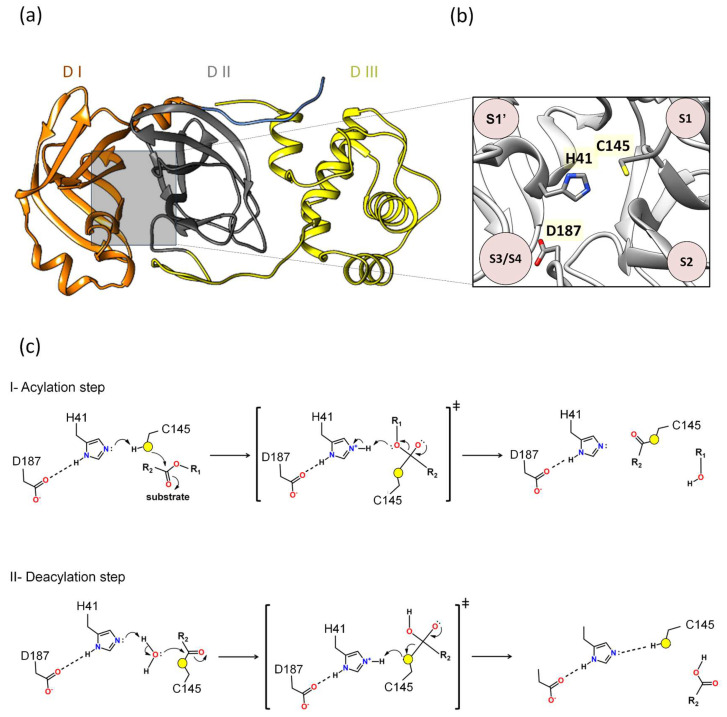
**Proposal of a novel mechanism of catalysis for 3CLpro:** (**a**) Domains I, II, and III are denoted as DI, DII, and DIII. (**b**) Between domains I and II lies the active site, where the proposed catalytic triad H41, C145, and D187 is located. (**c**) The proposed mechanism outlined in this review delineates two distinct phases: acylation (step I) followed by deacylation (step II), wherein the roles of specific residues mirror those of the catalytic triad observed in 3CLpro. In the acylation step, the active site of the main protease, Cys145 functions as a nucleophile. Initially, His41 serves as a base, taking a Cys145 proton, thereby activating it as a nucleophile. Next, the nucleophilic Cys145 initiates an attack on the carbon atom of the carbonyl group of the glutamine of the substrate, forming a tetrahedral intermediate. This enzymatic action leads to the cleavage of the peptide bond, resulting in the release of a substrate fragment with an alcohol group and the formation of an acyl bond between Cys145 and the residual fragment of the substrate. The deacylation step (step II) is the subsequent phase, where a water molecule assumes the role of a nucleophile and His41 functions as a base activating water as a nucleophile. Water makes a nucleophilic attack on the carbon atom of the acyl group bound to Cys145. Consequently, the acyl bond is cleaved, and the active form of the main protease is regenerated. Simultaneously, the second fragment with a carbonyl group of the substrate is released, thereby completing the enzymatic reaction. Asp187 may participate in modulating the p*K*_a_ of His41, allowing His41 to act as a base in the acylation and deacylation phases.

**Figure 3 viruses-16-00844-f003:**
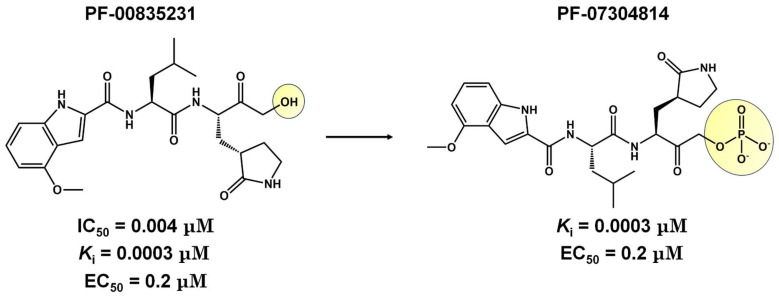
**Pharmacodynamics optimization for PF-00835231.** Biological activities (IC_50_ and/or *K*_i_ and EC_50_) are shown for PF-00835231 and PF–7304814.

**Figure 4 viruses-16-00844-f004:**
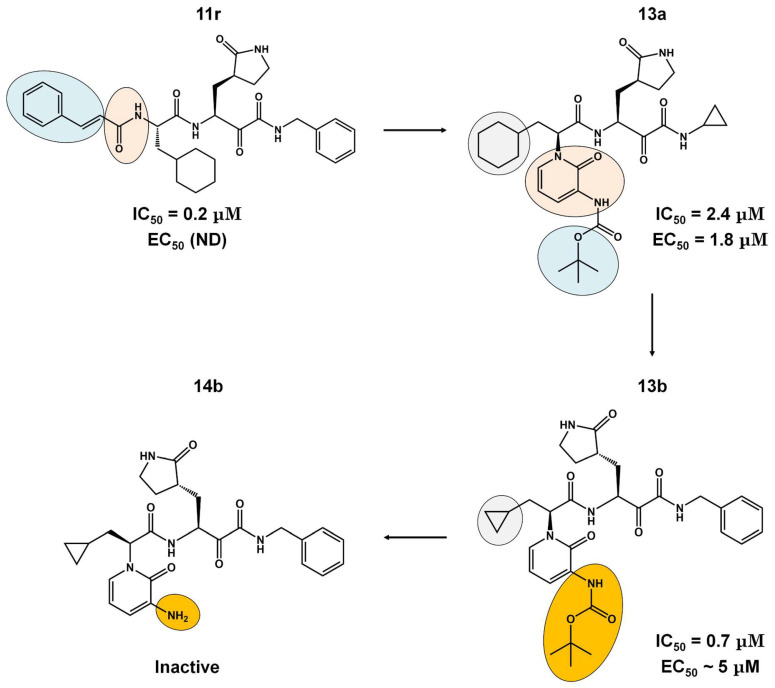
**Pharmacokinetic optimization for α-ketoamide derivatives.** Biological activities (IC_50_ and EC_50_) are shown for compounds **11r**, **13a**, **13b**, and **14b**. ND = not determined.

**Table 1 viruses-16-00844-t001:** **3CLpro inhibitors of SARS-CoV-2.** Column headings include the molecule name, structure, IC_50_, *K*_i_, CC_50_, EC_50_, and reference, representing the structural name, structural representation, half-maximal inhibitory concentration, inhibition constant, half-maximal cytotoxic concentration, half-maximal effective concentration against SARS-CoV-2 (that were propagated and tittered in Vero-E6 cells), and the respective reference for the provided data. The physical chemistry parameters MW, TPSA, HDA, HBD, and *consensus* Log*P* were estimated from the SwissADME website [85,86].

Inhibitor	Structure	IC_50_ (μM)	*K*_i_ (μM)	CC_50_ (μM)	EC_50_ (μM)	MW(Da)	TPSA(Å^2^)	HBA	HBD	Log*P*	Ref.
Nirmatrelvir (PF-07321332)	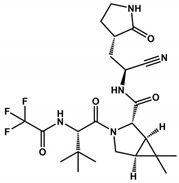	23.0	0.003	>10	0.08	499.5	131.4	8	3	1.9	[87,88]
GC376	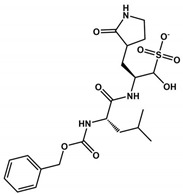	0.2	6.2	ND	2.2	484.5	182.3	8	4	0.7	[89]
1	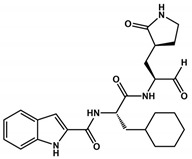	0.05	ND	>100	0.5	452.6	120.2	4	4	2.6	[30]
2	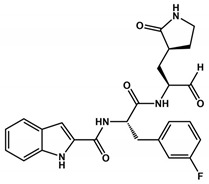	0.04	ND	>100	0.7	464.5	120.2	5	4	2.4	[30]
Boceprevir	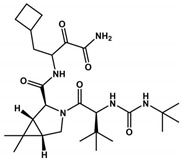	4.1	1.2	>100	1.3	519.7	150.7	5	4	2.1	[90]
PF-00835231	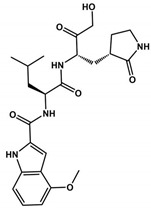	0.004	0.0003	ND	0.2	472.5	149.6	6	5	1.4	[75,91]
Ensitrelvir (S-217622)	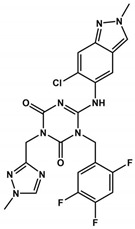	0.01	ND	>10	0.4	531.9	117.5	9	1	3.1	[92]
PF-07304814	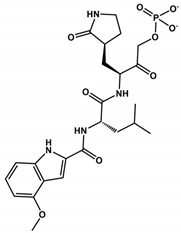	ND	0.0003	ND	0.2	550.5	211.6	9	4	0.7	[88]
Ritonavir	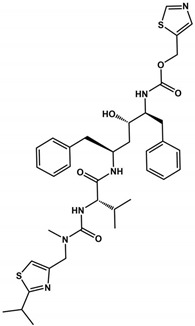	13.7	ND	94.7	19.9	720.9	202.3	7	4	5.0	[93,94]
Lopinavir	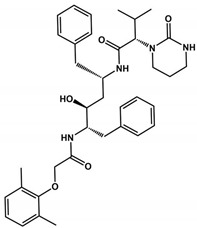	ND	ND	80.8	12.0	628.8	120.0	5	4	4.4	[93]

ND = not determined.

## Data Availability

Not applicable.

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
