# Peer review of "3-Chymotrypsin-like Protease (3CLpro) of SARS-CoV-2: Validation as a Molecular Target, Proposal of a Novel Catalytic Mechanism, and Inhibitors in Preclinical and Clinical Trials"

_viruses, 2024, doi:10.3390/v16060844_

Round 1

Reviewer 1 Report

Comments and Suggestions for Authors

This is a review 3CLprotease of SARS-CoV-2, discussing its catalytic mechanism and current inhibitors. As such, it is an important review.

My main suggestions are:

(1) SARS-CoV-2 should be in the title

(2) Novel results determined by the authors and presented for the first time here in this article should be presented in a section by themselves. Normally, a review article does not provide new information. Having a specific section with some title flagging that this is new information would help the reader know what is new and what is review.

(3) I'd suggest a table for inhibitors. Maybe one Figure with all of the structures. And then one table with the key characteristics of each inhibitor, as described in the text. That would be very useful, so the reader can tell at a glance what are the molecular weights, which ones are in clinical trial, what are their EC50s, etc. This information is in the text, at least some of it is. Difficult to know if it is all there, because it is introduced in prose. If the information is in table format, then the text can simply bring our attention to some key characteristics that the author would like to discuss. Which will be much more informative.

Reviewer 2 Report

Comments and Suggestions for Authors

In this review, the authors summarized the knowledge about 3CLpro as a drug target and presented the current state of coronavirus protease inhibitor testing. Overall, I enjoyed reading it and found it to be informative. While the article is of interest and up to date, I have some recommendations and criticisms aimed to make this review more thorough. 

-My main criticism is related to the bioinformatic analysis presented - it appears to be poorly thought-through (as explained below) and I have doubts if it contributes anything new beyond what has already been reported in the literature.

- Line 63 mentions 3CLpro cleaves 13 sites in ORF1ab, but this information seems incorrect. Also later on in the text (line 175) only 11 sites are mentioned, which is consistent with scientific literature.

 - It is unclear why the authors decided to make the link between overall sequence homology and inhibitor specificity, as known inhibitors target the catalytic sites or substrate binding sites only, and not the whole protein. Also, the sequence homology between CoV 3CLpro and identified human proteases is extremely low (Table S2) and in itself does not imply that 3Clpro inhibitors would be active against human proteases (if anything, it would suggest the opposite). This point should be made more clear in the text, instead of suggesting that these results demand experimental studies investigating inhibitor selectivity (line 115). 

- In line 140 it is mentioned that the structure of 3Clpro resembles that of pancreatic trypsin, yet this protein did not come up as any of the main hits in the bioinformatic analysis searching for structural homologues of 3CLpro. It is unclear why and I would appreciate if the authors would comment on this in the text. 

- I do not see how the authors arrived at only 3 positions being important in the catalytic activity (Figure S2) based on the presented alignment. There are multiple highly conserved positions apart from the 3 cited ones. Also, the sentence in the Figure legend: "The probability of these residues being important for the enzyme in the catalytic process based on phylogeny is 100%" makes no sense. The authors would need to show the probability calculations they performed to arrive at 100% probability number, which I doubt exist, and even if they did, phylogenetic analysis alone simply cannot ascribe a specific function to a conserved amino acid position. Thus this sentence needs to be changed or omitted.

- While citing IC50 and EC50 values for inhibitory compounds relevant primary references should be provided throughout the text. Also, it would be worth citing how these values were determined, as cell-free systems, virus-free cell line based assays and TCID50 infection-based assays tend to give different values. 

- The current evidence that Remdesivir (presented in line 303 and Fig. 4) acts through inhibition of 3CLpro is rather limited and mostly based on in silico analyses, thus calling it a potent 3CLpro inhibitor appears to be an overstatement. The widely accepted mode of action of this inhibitor is blocking of the viral RNA-dependent RNA polymerase, not protease. If the authors want to describe remdesivir as a 3CLpro inhibitor, the existing literature should be more critically discussed. 

- it might be unclear for some readers what Lipinski's rules are (line 366) thus I would suggest that the authors specify that high molecular weight compounds tend to have low bioavailability. 

- Table 1 title should be more specific, for example, "3CLpro inhibitors" rather than "Drugs"

- Authors should add information about ensitrelvir to Table 1

Comments on the Quality of English Language

I detected several language/spelling mistakes, for example:

- the plural form of genus is genera (incorrectly used in figure S1)

- Line 46:  should most likely start with " Until March 2024"

- lines 50 and 52 - no need to capitalize "Fusion" and "Internalization"

- Line 304 - should say "Figure" instead of "Figura"

-The sentence "Paxlovid is a novel oral administration that combines a 3CLpro inhibitor with Ritonavir or Paxlovid" (lines 351-354) makes no sense and is grammatically incorrect.

- Line 358 - there should be no accent above "Omicron"
